# RDET stacking classifier: a novel machine learning based approach for stroke prediction using imbalance data

Amjad Rehman[1], Teg Alam[2,3], Muhammad Mujahid[1], Faten S. Alamri[4], Bayan Al Ghofaily[1] and Tanzila Saba[1]

[1] Artificial Intelligence & Data Analytics Lab CCIS, Prince Sultan University, Riyadh, Saudi Arabia
[2] Department of Industrial Engineering, College of Engineering, Prince Sattam bin Abdulaziz University, Al Kharj, Saudi Arabia
[3] Azad Institute of Engineering and Technology, Azad Puram, Chandrawal *via* Bangla Bazar & Bijnour, Near CRPF Camp, Lucknow, India
[4] Department of Mathematical Sciences, College of Science, Princess Nourah bint Abdulrahman University, Riyadh, Saudi Arabia



Corresponding author
Faten S. Alamri,
fsalamri@pnu.edu.sa

## ABSTRACT

The main cause of stroke is the unexpected blockage of blood flow to the brain. The brain cells die if blood is not supplied to them, resulting in body disability. The timely identification of medical conditions ensures patients receive the necessary treatments and assistance. This early diagnosis plays a crucial role in managing symptoms effectively and enhancing the overall quality of life for individuals affected by the stroke. The research proposed an ensemble machine learning (ML) model that predicts brain stroke while reducing parameters and computational complexity. The dataset was obtained from an open-source website Kaggle and the total number of participants is 3,254. However, this dataset needs a significant class imbalance problem. To address this issue, we utilized Synthetic Minority Over-sampling Technique (SMOTE) and Adaptive Synthetic Sampling (ADAYSN), a technique for oversampling issues. The primary focus of this study centers around developing a stacking and voting approach that exhibits exceptional performance. We propose a stacking ensemble classifier that is more accurate and effective in predicting stroke disease in order to improve the classifier's performance and minimize overfitting problems. To create a final stronger classifier, the study used three tree-based ML classifiers. Hyperparameters are used to train and fine-tune the random forest (RF), decision tree (DT), and extra tree classifier (ETC), after which they were combined using a stacking classifier and a k-fold cross-validation technique. The effectiveness of this method is verified through the utilization of metrics such as accuracy, precision, recall, and F1-score. In addition, we utilized nine ML classifiers with Hyper-parameter tuning to predict the stroke and compare the effectiveness of Proposed approach with these classifiers. The experimental outcomes demonstrated the superior performance of the stacking classification method compared to other approaches. The stacking method achieved a remarkable accuracy of 100% as well as exceptional F1-score, precision, and recall score. The proposed approach demonstrates a higher rate of accurate predictions compared to previous techniques.

# INTRODUCTION

Stroke is an important cause of death, resulting in more than 6 million deaths annually across the world. A stroke has the potential to impact the parts of the brain responsible for regulating emotional responses, facilitating communication, and interpreting nonverbal cues in children (*Mendis, Davis & Norrving, 2015*; *Stroke Association, 2023*). It may also increase or decrease their sensitivity to sounds, contact, and other factors. A stroke is a sudden neurological condition affecting the blood vessels in the brain. It arises when the blood flow to a specific brain region is interrupted, leading to oxygen deprivation to the brain cells. The cerebrum, a vital central nervous system component, is responsible for governing various physiological processes including memory, movement, cognition, speech, and the autonomic regulation of essential organs. Nevertheless, the outcomes can be potentially life-threatening. Prompt medical attention is crucial in addressing a stroke, an acute condition demanding immediate intervention (*Katan & Luft, 2018*).

Stroke impacts not only the individual directly but it has an impact on those closest to the patient. Further, considering why many individuals suspect, stroke can affect anyone, regardless of their age, physical health and gender (*Elloker & Rhoda, 2018*). It is categorized into two types: ischemic and hemorrhagic. The severity of stroke can range from mild to extremely severe. Hemorrhagic strokes cause cerebral hemorrhage when a blood artery within the brain bursts. However, ischemic strokes, that's are more prevalent, occur when an arterial blockage-blood flow to specific regions of the brain (*Bustamante et al., 2021*). According to the World Stroke Organization (*Rohit et al., 2022*), approximately 13 million people experience a stroke each year, leading to an estimated 5.5 million fatalities. Cerebral strokes rank as the sixth leading cause of fatality in the U.S and the 4th leading factor of death in India. In the United States, around 795,000 cases befall each year, continuingly resulting in lifelong impairment (*Cowan et al., 2023*). The India collaborative acute stroke-study (ICASS) report reveals that over 2,000 individuals in India suffered from stroke (*Banerjee & Das, 2016*). Meanwhile, in Canada, the overall stroke mortality rate surpassed 15,000 cases in the year 2000 (*Alruily et al., 2023*). Stroke stands is the primary cause of death and disability worldwide, making its impact profound across all facets of life.

Several factors contribute to the plausibility of experiencing a stroke, including a history of previous stroke, transient stroke, and other heart conditions. Age also plays a role, with individuals over 55 years of age being at higher risk. Furthermore, stroke exhibits a rapid progression, and its symptoms can manifest in various ways. Sometimes, symptoms may develop gradually, while they can emerge suddenly in other instances. Remarkably, it is even plausible for individuals to wake up from sleep already experiencing symptoms. The primary indications include arm or leg paralysis, aches in the limbs or face, speech difficulties, impaired walking ability, dizziness, reduced vision, headache, vomiting, and a drooping mouth (facial asymmetry). In severe stroke cases, the patient may lose consciousness and enter a coma state (*Mosley et al., 2007*; *Lecouturier et al., 2010*). Diagnostic tests such as carotid triplex and cardiac triplex may be employed. Strokes can range from severe (extensive) to mild, with the initial 24 h playing a critical role in most cases. The treatment approach, typically pharmaceutical but occasionally surgical, is

determined based on the diagnosis. In cases where the patient has collapsed addicted to a coma, intubation and automated freshening become necessary in ICU (*Gibson & Whiteley, 2013*; *Rudd et al., 2016*). After a stroke, some patients recover, but depending on the severity of the stroke, most people are still struggling. There may be issues with speaking or understanding-speech, as well as issues with concentration or memory, cognitive issues, emotional issues such as depression, loss of balance or mobility, sensory loss on one side of the body, and difficulty swallowing food (*Delpont et al., 2018*; *Kumar, Selim & Caplan, 2010*).

The acute phase of a stroke is thereafter linked to enduring cellular damage, which serves to diminish cellular plasticity and regenerative capacity. The selection of treatment techniques is predicated upon these distinct phases, with the aim of optimizing neuronal preservation and rehabilitation. To minimize the risk of experiencing a stroke, it is essential to adopt several preventive measures. *Akter et al. (2022)* presented a model that incorporated an algorithm for achieving accurate estimates of brain stroke. Effective approaches for data collection, data pre-processing, and data transformation have been employed in order to ensure the reliability of the data used in the proposed model. The model was developed using a dataset called brain stroke prediction. The training and testing approach involves the use of three classifiers; RF, SVM, and DTs. A number of performance evaluation metrics, including accuracy, sensitivity, error rate, false-positive rate, false-negative rate, root mean square error, and log loss, were used to evaluate the performance. These include regularly monitoring blood pressure, maintaining a healthy weight, quitting smoking and excessive alcohol consumption, and adhering to a balanced diet low in fat and salt (*Pandian et al., 2018*; *Almadhor et al., 2023*). In *Hung et al. (2017)*, the authors presented an electronic medical record dataset and compare the results of deep neural networks with ML for stroke classification. Deep NN and RF models demonstrate high accuracy to other ML algorithms. In *Liu, Fan & Wu (2019)*, a hybrid ML approach was developed for stroke prediction. In *Ong et al. (2020)*, a comprehensive framework was introduced, utilizing machine learning and natural language processing (NLP), to determine the severity of ischemic stroke from radio graphic text. The application of the NLP algorithm exhibited superior performance compared to other algorithms. *Al Duhayyim et al. (2023)* used ensemble learning for the prognosis of stroke. The low prediction accuracy and Imbalance stroke dataset issues could have been studied better in the past.

## Research motivation

Balanced datasets were an issue in past research on brain stroke predictions utilizing stroke datasets; however, few medical stroke datasets are capable of replicating such standards with less accurate findings. In the past, the researchers used high-performance models on imbalanced data to achieve maximum accuracy. Only minority classes receive high accuracy from the imbalanced dataset, which delivers low accuracy to all minority classes. Predictions are inaccurate as a result of the extreme imbalance in some datasets. Almost all medical-related disease datasets suffer from Imbalanced data. To balance the dataset and lower the feature dimensions, several authors used feature reduction and undersampling

approaches. When researchers use undersampling approaches on medical datasets, they could end up with less accuracy and the loss of crucial information from the majority classes. This study employed two important oversampling techniques, SMOTE and ADASYN, that addressed the imbalanced dataset problems. The authors in this study proposed a voting ensemble model that minimized false positive and negative rates and achieved outclass performance. The proposed model was also investigated with the K-fold validation methodology, which provides better results than previous methods. The results of this study help clinicians identify brain strokes early and effectively. The proposed approach presents great significance in addressing the risk of stroke since individuals experiencing memory impairments have challenges in cognitive functioning and decision-making abilities in a social context.

## Main contributions

The majority of the models utilized in previous studies were those from machine learning, and ensemble models for stroke prediction are still understudied. The low prediction accuracy and imbalanced stroke dataset issues could have been studied better in the past. A comprehensive framework is needed to determine the severity of the stroke from stroke datasets. The main contributions of our study are as follows:

- A novel, high-performance RDET stacking classifier approach is proposed for stroke prediction in its early stages and performed more extensive experiments than previous studies. A stacking RDET classifier is an efficient approach for identifying individuals with a high long-term risk of suffering a stroke.
- The most significant ADASYN and SMOTE Techniques are used to balance the heavily Imbalanced dataset, and their performance on single and ensemble ML classifiers is investigated. The experiments demonstrated the stacking method's effectiveness as opposed to single models and voting classifiers, with exceptionally high accuracy.
- We fine-tuned nine ML classifiers to make predictions on imbalanced and balanced datasets. Compare stacking ensemble and soft voting ensemble classifiers with single ML classifiers.
- The proposed stacking ensemble model has been evaluated using cross-experiments with k-fold validation, and statistical tests are performed to compare it to other models.

## LITERATURE REVIEW

Over the past few years, the research community has witnessed a remarkable increase in interest surrounding the advancement of tools and techniques to monitor and forecast diseases that significantly affect human health. In this section, we delve into the most recent progress made in harnessing the influence of machine learning (ML) methods to forecast the risk of stroke. Particularly noteworthy is the successful integration of ML approaches, which have demonstrated significant potential in providing more precise predictions of stroke outcomes when compared to conventional methods.

In *Xie et al. (2021)* researchers examined the application of Artificial Intelligence (AI) techniques for predicting strokes. The study employed an innovative method by utilizing

the decision tree (DT) algorithm with principal component analysis (PCA) for feature extraction. To construct the predictive model, the researchers employed a neural network classification algorithm, which resulted in an impressive accuracy rate of 97% when tested on the Cardiovascular Health Study (CHS) dataset. A study (*Adi et al., 2021*) conducted recently showcased the successful utilization of various ML techniques in predicting individuals at a high-risk of stroke. The investigation incorporated three ML algorithms: random forest (RF), DT, and naive Bayes (NB). By assessing each approach, predictions were made based on the patient's medical histories as attributes within the respective models. The RF technique demonstrated the highest accuracy of 94.781% among the methods employed.

In *Dritsas & Trigka (2022)*, the authors examined the effectiveness of various ML models in accurately predicting stroke cases using participant profiles. The stacking classification algorithm demonstrated remarkable accuracy by incorporating various elements from the participant profiles, surpassing 98%. This highlights the stacking technique as an effective method for identifying patients at high risk of experiencing a stroke. Additionally, in *Tazin et al. (2021)*, the authors trained four distinct models using ML algorithms and multiple physiological parameters to predict strokes effectively. Among these models, the RF model exhibited exceptional performance, achieving an accuracy rate of approximately 96%. These findings underscore the effectiveness of the RF model in accurately predicting strokes.

In a recent study (*Govindarajan et al., 2020*), researchers conducted a study on stroke disorders classification using a combination of text-mining techniques and ML classifiers. The study involved data collection from 507 patients, and various ML approaches, including Artificial Neural Networks (ANN), were employed for training. Among these approaches, the SGD algorithm exhibited the highest performance, achieving an impressive accuracy of 95%. In another study (*Amini et al., 2013*; *Reza, Rahman & Al Mamun, 2014*), researchers aimed to predict stroke incidence by examining a cohort of 807 subjects. This cohort comprised healthy individuals and individuals with specific health conditions associated with stroke risks. The researchers utilized two techniques, namely the c4.5 DT algorithm and the KNN algorithm. The c4.5 DT algorithm demonstrated an accuracy of 95%, while the KNN algorithm achieved a slightly lower accuracy of 94%. These two techniques emerged as the top-performing methods in their analysis. Additionally, in a separate publication (*Bulygin et al., 2020*), researchers reported on estimating the prognosis for ischemic stroke. They utilized data from 82 patients diagnosed with ischemic stroke and employed two ANN models to assess precision. The models achieved precision rates of 79% and 95% respectively, highlighting their effectiveness in predicting the prognosis of ischemic stroke.

In recent research (*Sung et al., 2015*), the objective was to develop a stroke severity index. The researchers gathered data from 3,577 patients who had experienced acute ischemic stroke. They adopted different data mining algorithms, including linear regression, to construct classification models. Among these techniques, the KNN model exhibited the most promising outcome for predicting stroke severity, with a 95% Confidence Interval (CI). In *Monteiro et al. (2018)*, the focus was on predicting the

functional outcome of ischemic stroke using ML. This approach was applied to patients who had been admitted three months prior. The results indicated that the ML method had an AUC value exceeding 90%, indicating its effectiveness in predicting functional outcomes. In *Kansadub et al. (2015)*, researchers investigated to forecast stroke risk using different ML algorithms such as NB, DT, and NN. Accuracy and AUC were used as assessment metrics. The DT and NB classifiers demonstrated the highest accuracy among the tested algorithms in predicting stroke risk. Furthermore, in *Adam, Yousif & Bashir, (2016)*, researchers conducted a study on classifying ischemic stroke. They employed two models, namely the KNN and DT algorithms, for classification purposes. Based on their research findings, the DT algorithm proved to be more practical for medical specialists in accurately classifying strokes. In *Pradeepa et al. (2020)*, a methodology was proposed to identify different stroke symptoms and preventive measures using social media resources. The authors developed an architecture that employed spectral clustering to iteratively group tweets based on their content. The PNN algorithm outperformed the others, and achieved 89.90% accuracy.

*Saini, Guleria & Sharma (2023)* employed four main classifiers in their study, including naive bayes, kstar, multilayer perceptron, and random forest. The classifiers utilized in this study were trained using the Kaggle brain stroke dataset. The performance of these classifiers was evaluated using the WEKA method, with metrics such as accuracy, recall, precision, f-measure, and running time being considered. The proposed work revealed that random forests exhibited the highest performance, with an overall accuracy of 95%. Furthermore, a comparative analysis has been conducted to evaluate the performance of the proposed model in relation to previously published research, revealing that the novel approach exhibits superior performance. The potential implications of aiding physicians in identifying potential cases of stroke extend to the healthcare sector as well. *Kumari & Garg (2023)* also utilized multiple classifiers based on machine learning for categorizing patients with stroke. These classifiers were subsequently compared to one another. The researchers utilized local interpret-able model agnostic- explanation and SHAP to elucidate the reasoning behind the decision made by the most effective machine learning model. The findings indicate that RF yields the most accurate predictions.

An open access dataset was utilized to predict the probability of a stroke using machine learning techniques, namely the random forest, extreme gradient boosting, and light-gradient boosting algorithms. The stroke prediction dataset was subjected to pre-processing techniques, which encompassed the use of the K-nearest neighbors (K-NNs) imputation method to handle missing values, the removal of outliers, the utilization of one hot-encoding for categorical variables, and the normalization of features with disparate value ranges. The technique of synthetic minority oversampling (SMO) was employed in order to achieve a balanced representation of stroke dataset. Furthermore, a random search technique was employed to optimize the hyper-parameters of the machine learning algorithm, aiming to get the most optimal parameter values. After employing the tuning strategy, the researchers proceeded to analyze and compare its performance with that of traditional classifiers. A high level of accuracy, namely 96%, was seen in their *Alruily et al. (2023)* study. Also, *Rahman, Hasan & Sarkar (2023)* used deep learning and machine

learning to predict early-stage brain strokes. The methodology was evaluated using a stroke prediction dataset from Kaggle. This study used extreme gradient boosting, AdaBoost, light gradient boosting machine, random forest, decision tree, logistic regression, k neighbors, naive Bayes, and deep neural networks for the complete classification tasks. The random forest classifier outperforms other machine learning classifiers with remarkable accuracy. The four layer neural network algorithm outperforms the three-layer ANN while using the same features. Machine learning algorithms outperformed deep neural networks. An ensemble model was constructed by *Premisha et al. (2022)* from base, bagging, and boosting strategies. To see the effects of tuning, the models are implemented in the training phase with and without hyperparameter tuning. The ensemble made with the voting classifier achieved the highest results.

Moreover, a study conducted by researchers (*Sailasya & Kumari, 2021*) employed the Kaggle dataset. The primary focus of the research was to implement various ML algorithms, such as LR, DT, RF, KNN, SVM, and NB. Notably, the NB algorithm demonstrated remarkable accuracy, achieving 82% for stroke prediction. Furthermore, a separate analysis (*Nwosu et al., 2019*) was conducted on patients EHR's (electronic health records) to assess risk factor on stroke prognosis. After performing 1,000 experiments with the EHR dataset, the NN, DT, and RF classifiers achieved classification accuracies of 75.02%, 74.31%, and 74.53%, respectively. For the diagnosis of various diseases, researchers such as *Amin et al. (2019)*, *Saba et al. (2018)*, *Saba, Rehman & Sulong (2010)*, *Alsubai et al. (2022b, 2022a)* and *Abunadi & Senan (2022)* have conducted studies on health-related diseases. In their study (*Abunadi et al., 2022*), the researchers introduced an ML approach that utilized 206 clinical variables. Their method yielded impressive results. To extract the most important information for correct diagnosis, a widely recognized AI technology, is employed. This involves techniques such as Text vectorization and Dl algorithms. The related work summary is presented in Table 1.

# PROPOSED STROKE PREDICTION METHODOLOGY

The proposed stroke prediction methodology is presented in Fig. 1. We obtained a stroke prediction dataset from Kaggle, which has 11 features. Preprocessing is performed to handle missing values and then normalizes the dataset to improve performance and robustness. After that, the imbalanced dataset is balanced with ADASYN and SMOTE oversampling techniques. Then training, testing, model implementation, stacking and voting ensemble classifiers, and model evaluation are described briefly in the subsection.

### Dataset description

The dataset was obtained from an open-source Kaggle website (*Kaggle, 2023*). There are 5,110 rows, 11 features, and 3,254 participants. A total of 10 features are given to the proposed model as input and one feature is referred to the target class as described as follows:

- **Gender:** This feature shows the gender of the participant. There is a count of 1,260 men and 1,994 women.

**Table 1 Summary of related work.**

| Techniques | Dataset | Performance metrics | Advantages | Disadvantages |
|---|---|---|---|---|
| *Adi et al. (2021)* (RF, DT, NB) | Stroke prediction dataset | Accuracy, precision, recall and f1 score | Three machine learning models were utilized in this work to predict strokes using patient history as an input. The maximum accuracy was attained by the RF, at 94.7%. | The study does not discuss the preprocessing techniques used to normalize and prepare the dataset. In addition, the stroke dataset was imbalanced, and the techniques used were not specified. The accuracy was inadequate. |
| *Dritsas & Trigka (2022)* (KNN, NB, LR, SGD, MLP, Stacking) | Stroke prediction dataset | Accuracy, precision, recall and f1 score, AUC | The authors conducted preprocessing on the stroke dataset and employed the Synthetic Minority Over-sampling Technique (SMOTE) to address class imbalance. | The findings obtained are unsatisfactory. |
| *Tazin et al. (2021)* (RF, LR, DT, Voting classifier) | Stroke prediction dataset | Accuracy, precision, recall and f1 score | The researchers conducted many preprocessing techniques to enhance the reliability and balance of the dataset. Additionally, a voting classifier was employed in order to improve the outcomes. The confusion matrix and report were also prepared to display the findings for each class. | The study failed to implement cross-dataset experiments, and the utilization of a voting classifier yielded less precise results. |
| *Govindarajan et al. (2020)* (ANN, SVM, Bagging, Boosting, RF) | Stroke prediction dataset | Accuracy, precision, recall and f1 score, STD | The authors used data case sheets from 507 patients and obtained an accuracy score of 95%. | The dataset that was obtained was relatively small, and the results were unsatisfactory. |
| *Amini et al. (2013)* (DT, KNN) | Stroke prediction dataset | Accuracy | The researchers achieved a 95.4% accuracy rate by employing a sample size of 807 individuals, comprising both healthy and sick individuals. Data was obtained using a standardized checklist consisting of 50 risk variables associated with stroke, including features such as a prior history of cardiovascular disease. | The researchers did not employ ML techniques effectively, nor did they specify the criteria that were used to choose the algorithms that yield good results. |
| *Monteiro et al. (2018)* (LR, XGBoost, RF, SVM) | Ischemic stroke patients | Accuracy, AUC | Machine learning was used to predict ischemic stroke patients' three-month functional outcomes with an accuracy of 88%. However, when features were gradually added, the area under the curve increased above 0.90. | Their stroke prediction accuracy was very low. |
| *Li et al. (2019)* (NB, BN, LR, DT, RT) | National stroke screening data | Accuracy, precision, recall and f1 score | The training set imbalance is fixed by employing oversampling and undersampling techniques. The degrees of stroke risk are then assessed using a range of classification models | They used a variety of ML models, but their detection results were poor. |
| *Sailasya & Kumari (2021)* (KNN, SVM, DT, NB, LR) | Stroke prediction dataset | Accuracy, precision, recall and f1 score, | The authors employed 5 distinct machine classifiers to predict a brain stroke. The under sampling approach is used to balance the severely imbalanced dataset. | The undersampling strategy in use might remove important features from the majority class that result in inaccurate prediction. |
| *Nwosu et al. (2019)* (DT, RF, NN) | Electronic health dataset | Accuracy | To determine the influence of risk variables on stroke prediction, they examine patients' electronic medical data. | There was no comparison with previous efforts or existing research, and the accuracy achieved was not very high. Additionally, k-fold cross-validation is not used in the analysis of the results. |

| Techniques | Dataset | Performance metrics | Advantages | Disadvantages |
|---|---|---|---|---|
| *Ong et al. (2020)* (KNN, CART, OCT, RNN) | MRI health reports | Accuracy, AUC, Specificity, Sensitivity, Threshed | The researcher's utilized magnetic resonance imaging case reports as a means of identifying and diagnosing ischemia conditions. The Bag-of-Words (BoW), Global Vectors for Word Representation (GloVe), and Recurrent Neural Network (RNN) are employed. | The study does not pertain to stroke prediction and is limited to the analysis of MRI records from only two hospitals. The attained accuracy of the study is quite inadequate. |

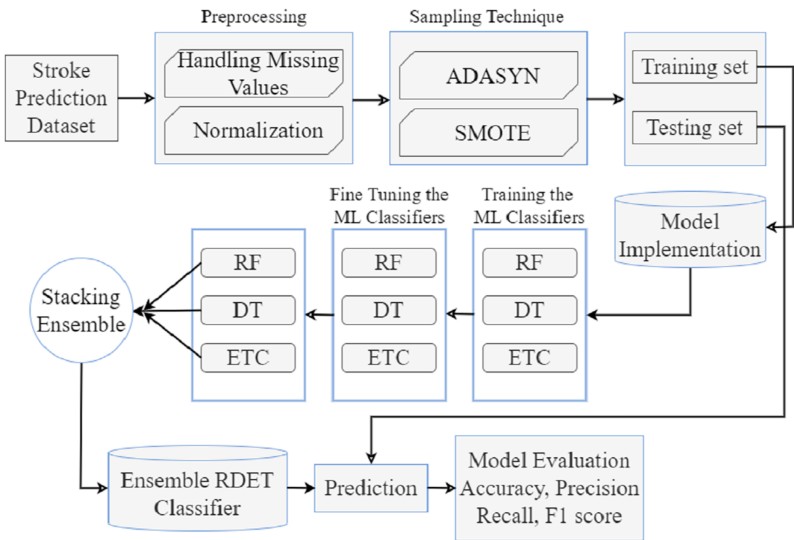

**Figure 1 Proposed stroke prediction methodology diagram.**

- **Age:** This feature refers to participant's age who are above 18 years.
- **Hypertension:** This feature shows whether the participant is suffering from hypertension. Participants with hypertension have a percentage of 12.54% in the dataset.
- **Heart disease:** This feature shows if the participant is a patient with heart disease or not.
- **Ever married:** The married individuals are 79.94% in this feature.
- **Work type:** This shows the work type of the participants and it consists of four categories; private (65.02%), self-employed (19.21%), government jobs (15.67%), and never worked (0.1%).
- **Residence type:** This feature shows whether the individual lives in an urban or rural area. It has two categories; urban (51.14%) and rural (48.86%).
- **Average glucose level (mg/dL):** This variable measures the participants glucose level.
- **BMI (kg/m$^2$):** This feature measures the participants body mass-index.
- **Smoking status:** This attribute shows whether the user smokes or not.
- **Stroke:** This feature shows whether an individual has a history of stroke or not. The number of individuals who have experience a stroke is 5.53%.

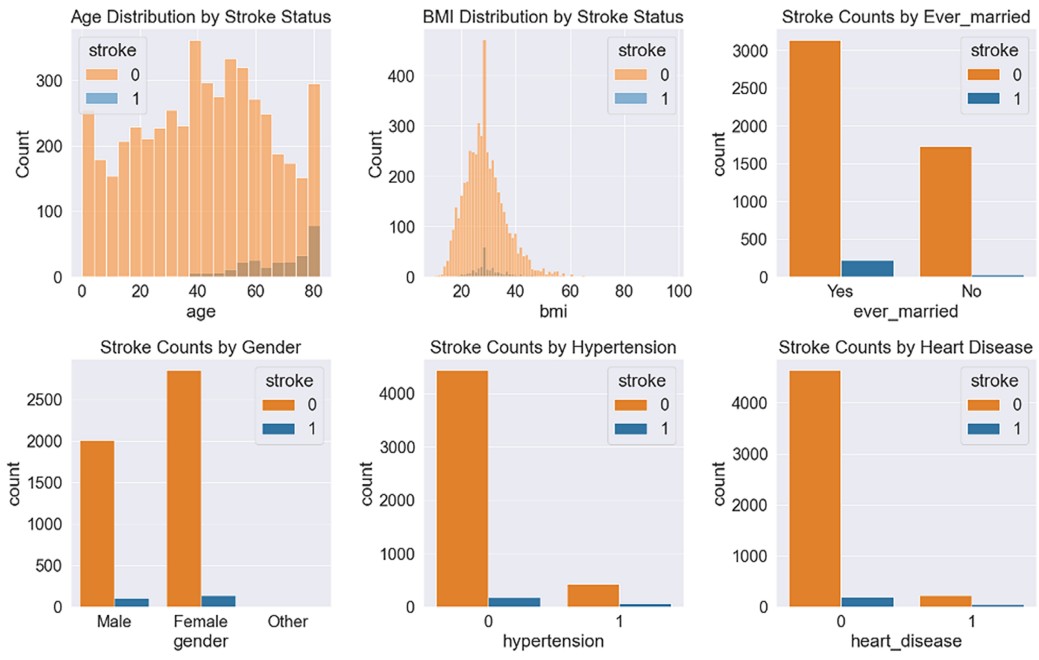

**Figure 2 Some significant features obtained from the stroke prediction dataset.**

The majority of attributes in the dataset are categorical. The dataset includes 2,994 females, 2,115 males, and one individual that is labelled as "other." It includes 4,861 individuals classified as normal (healthy) and 249 individuals with a stroke (*Shah & Cole, 2010*; *Howard, 2021*; *Lee et al., 2020*).

## SMOTE and ADAYSN oversampling techniques

The Synthetic Minority Oversampling Technique (SMOTE) and Adaptive Synthetic Sampling (ADAYSN) (*He et al., 2008*) are most significant techniques used in ML to address the class imbalance problems in dataset. The problem of class imbalance arises when the count of samples in one class considerably outweighs the count of samples in the other class, leading to a biased model. SMOTE (*Mujahid et al., 2021*) works by creating synthetic samples from the minority class to balance the dataset. It does this by recognizing the minority class samples and developing new synthetic samples along the line segments between the feature space of existing minority class samples. This method helps to increment the minority class representation, thus handing a more balanced training set for the model. By moderating the effects of class imbalance, SMOTE improves the accomplishment and perfection of ML models, specifically in scenarios where minority class samples are crucial but scarce.

After balancing the stroke dataset, we divide the dataset into two sets: train and test. A total of 80% of the data is employed for training the classifiers and 20% for testing their performance. The most important features from the stroke prediction dataset are displayed in Fig. 2 using plots.

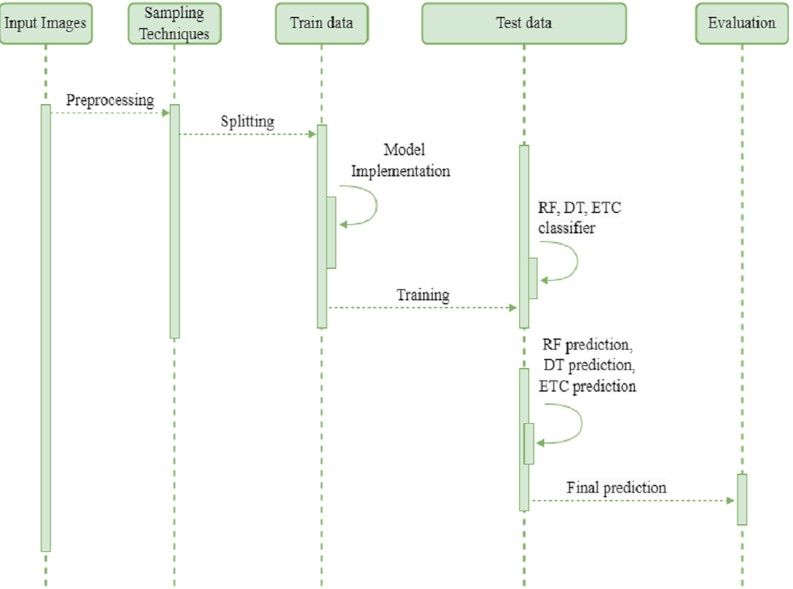

**Figure 3 Sequence diagram of proposed method.**

## RDET stacking classifier

Stacking is an ensemble method for machine learning that uses multiple models to make a powerful model (*Rajagopal, Kundapur & Hareesha, 2020*). Using cross-validation methods like k-fold cross validation, each model is trained on multiple sub-sets of the data. The predictions from each model are then added together to get the final forecast. This method often leads to better performance because the different models can learn things that each other does not know. Stacking can decrease the difference between predictions, which makes it a useful method for datasets that need to be balanced. Stacking can also combine different kinds of models, like neural networks and decision trees. Stacking is a more complex way to use machine learning than other methods, so the different models and how they work together must be fine-tuned. It can help make your model better at making predictions. Overfitting may also be decreased by stacking. Training each classifier on a different subset of data, prevents from training overfitting the model. When we use a complex ensemble method like boosting or deep learning, it can be hard to understand how the final results are generated. But if we combine a number of smaller classifiers, to understand the final predictions.

We employed three tree based Ml classifiers to make a final single classifier. The random forest (RF), decision tree (DT), and extra tree classifier (ETC) are first trained and fine-tuned with hyperparameters. Then combined through a stacking classifier with a k-fold cross validation methodology. With k-fold, the ensemble classifier learns different increasing training subsets more efficiently, which may help increase. The ensemble stacking classifier makes predictions on new or unseen data. The effectiveness of the proposed ensemble classifier is evaluated through various performance metrics. The sequence diagram of the proposed method presented in Fig. 3. The input images endure preprocessing and preprocessed data are balanced using oversampling techniques such as

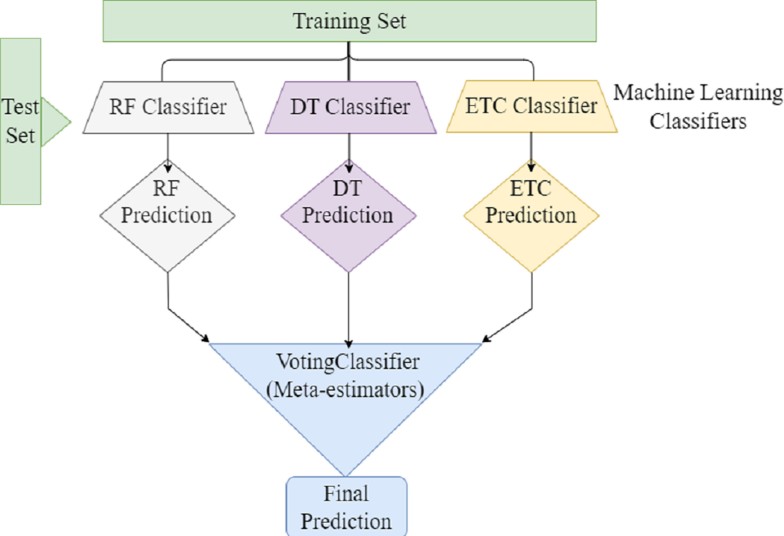

**Figure 4 Ensemble voting ML classifier.**

ADASYN and SMOTE. Next, the oversampled data is divided into training and testing sets. During the training phase, the authors implemented the development of a model. Subsequently, they trained random forest (RF), decision tree (DT), and extremely randomized trees (ETC) classifiers. Finally, they employed the stacking approach to combine the predictions generated by these classifiers. The final prediction is assessed using four basic metrics.

## Ensemble soft voting classifier

Ensemble learning can be used to address the several challenges faced by machine learning. To develop a classification model. The ensemble learning technique combines different machine learning classifiers to develop a classification model. Ensemble learning combines multiple classification models rather than relying on single models to increase the model's performance. The efficacy of classification can be increased by combining the predictions of multiple machine learners compared to a single machine learner. Ensemble voting classifiers have the potential to reduce prediction bias- and -variance. By combining multiple classifiers, we can compensate the weaknesses of single models and generate more accurate prediction model. Furthermore, Ensemble Voting can assist in addressing class Imbalance problems by giving extra importance to classifiers who perform outclass in minority classes. In soft voting, each model provides a probability of belonging to a certain class based on specific data features. Furthermore, each prediction the model makes is weighted according to its importance. After that, the class with the highest sum of weighted probabilities is assigned to the data. Figure 4 presents the voting classifier that combined three ML classifiers (RF, DT, and ETC), and predictions through meta-estimators or voting classifiers, to produce the final prediction.

**Table 2 Parameters tuning for machine learning.**

| Model | Hyper parameters |
| --- | --- |
| SVM | kernel='sigmoid', C=2.0, random_state=100, probability=True |
| DT | random_state=50, max_depth=100 |
| ETC | n_estimators=100, random_state=150, max_samples=0.5, max_depth=50, bootstrap=True |
| KNN | n_neighbors=3 |
| GBM | n_estimators=100, random_state=50, max_depth=100 |
| SGD | loss="modified_huber", penalty="l2", max_iter=5 |
| RF | n_estimators=10, random_state=150, max_samples=0.5, max_depth=50 |
| LR | random_state=300, solver='newton-cg', multi_class='multinomial', C=1.0 |
| ADA | n_estimators=100, random_state=50 |

## Machine learning

Machine learning (ML) models that can automatically learn and improve from experience without being explicitly programmed. In this section, we introduce the ML models employed in the classification framework to predict stroke risk. We will utilize a range of classifiers. The hyper parameters and its tuning values are presented in Table 2.

### Decision tree

A decision tree (DT) is an ML algorithm that uses a tree like model to make predictions or decisions based on input features (*Santos et al., 2022*). It is a graphical representation where each 'internal node' represents a feature each branch represents a decision rule, and each leaf node represents a class. However, they can be prone to overfitting and high variance. DT are versatile and interpretable ML algorithms that handle classification and regression tasks. While they have certain limitations, they serve as a fundamental building block in many advanced algorithms and have proven to be valuable in many applications.

### Extra trees classifier

The Extra Trees classifier (ETC) is an ML algorithm that variant of the RF algorithm. Like RF, Extra Trees constructs an ensemble of DTs to make predictions. However, what sets Extra Trees apart is its higher level of randomization during the tree construction process. This randomization leads to greater diversity among the DT, making Extra Trees more robust to noise and reducing overfitting. Additionally, Extra Trees can be computationally faster than RF as it does not require calculating the best split at each node. Extra Trees are often used for classification tasks and can handle numerical and categorical features. It can handle high-dimensional datasets, provide feature importance rankings, and deliver good generalization performance.

### Support vector machine

A support vector machine (SVM) is a supervised model that works to solve classification problems. An algorithm-generated hyperplane separates the data into two distinct categories. The prevailing opinion holds that the optimal condition is a hyperplane with the greatest practicable margin between classes. Those above the hyperplane are designated

as first class, while those below the hyperplane are designated as second class (*Zhou et al., 2022*). The performance of SVM in NLP, data mining, image processing and for other classification tasks is outstanding.

### K-nearest neighbors

K nearest neighbors (K-NNs) classifier operates on finding the K nearest data points in the feature space to a given query point and making predictions based on the majority vote. KNN does not require any training process as it stores all the training data points in memory. It is a non-parametric algorithm, meaning it makes no assumptions about the underlying data distribution. KNN is easy to implement and understand. Therefore, it is computationally expensive, especially for large datasets, as it requires calculating distances between data points (*Yean et al., 2018*).

### Gradient boosting machine

The gradient boosting machine (GBM) is a supervised ML algorithm that can be used to make predictions or classify data. It works by building a series of decision trees, where each subsequent tree is trained to correct the mistakes of the preceding tree. This process continues until the model reaches a certain level of accuracy or a specified number of trees have been built. By iteratively improving the model's performance, GBM constructs a strong ensemble model that can make accurate predictions. GBM is known for its ability to handle complex relationships and capture nonlinear patterns in the data (*Islam, Debnath & Palash, 2021*). However, GBM can be prone to overfitting, so regularization techniques such as learning rate adjustment, tree depth limitation, and early stopping are often employed to prevent this.

### Random forest

Random forest (RF) is a popular ML algorithm that combines the power of DTs and ensemble learning (*Dritsas & Trigka, 2022*). It constructs multiple DTs and then combines their predictions to make a final prediction. Each DT in the random forest is trained on a random-subset of the training data and a random subset of features. This randomization helps to reduce overfitting and increase the model's generalisation ability. Random forest is known for its, scalability, robustness and ability to address high dimensional data.

### Logistic regression

Logistic regression (RF) is a widely used machine learning algorithm for binary classification tasks (*GholamAzad et al., 2022*). Despite its name, it is a regression algorithm that models the probability of an instance belonging to a particular class. It estimates the parameters of a logistic function, also known as the "sigmoid function", which maps the input features to a value between 0 and 1. This value represents the probability of the instance belonging to the positive class. During training, the algorithm optimizes the parameters using various optimization techniques. Logistic regression is widely used in various domains, including healthcare, finance, and social sciences, due to its simplicity, interpretability, and effectiveness in handling binary classification problems.

### Adaptive boosting

Adaptive boosting (ADA) (*Islam, Debnath & Palash, 2021*) is an ML algorithm that used for classification and regression problems. It combines different "weak" models to create a single "powerful" model. Each iteration of the algorithm assigns higher-weights to misclassified data points and lower points to correctly classified data points. This process continues until the model reaches a certain level of accuracy or a specified number of iterations have been completed. AdaBoost is known for its ability to handle complex datasets and effectively deal with high-dimensional feature spaces. It is particularly effective in boosting the performance of decision trees, creating a boosted version called AdaBoost decision trees (AdaBoostDT) or simply AdaBoost. AdaBoost is widely used in various applications, including face detection, text classification, and object recognition, natural language processing, due to its flexibility, robustness, and ability to handle large-scale datasets.

## Evaluation metrics

Evaluation metrics are numerical indicators employed to evaluate the effectiveness of a model or system in addressing a particular task (*Abunadi, 2022*). The classification outcomes produced by the model can be categorized into four groups: true positives (TP), true negatives (TN), false positives (FP), and false negatives (FN). TP denotes correctly identified positive instances, while TN represents accurately identified negative instances. FP signifies incorrectly predicted positive instances, and FN represents incorrectly predicted negative instances. Several evaluation parameters have been employed in these studies, including recall, precision, accuracy, AUC, and F1 score.

### Accuracy

Accuracy is a metric that quantifies the frequency with which a model accurately predicts the outcome or class of a given sample.

$$Accuracy = \left( \frac{TP + TN}{TP + FN + FP + TN} \right) \tag{1}$$

### Precision

Precision is a metric that evaluates the ratio of correctly predicted positive samples (known as true positives) to the total number of positive predictions generated by the model.

$$Precision = \left( \frac{TP}{TP + FP} \right) \tag{2}$$

### Recall

Recall is a metric that quantifies the ratio of correctly identified positive cases (referred to as true positives) to the sum of true positives and false negatives. It measures the model's ability to accurately identify actual positive instances.

$$Recall = \left( \frac{TP}{TP + FN} \right) \qquad (3)$$

### F1-score

The F1-score is a commonly employed performance metric for binary classification tasks, which merges both precision and recall. It is calculated as the harmonic mean of precision and recall, resulting in a single value that represents their balanced combination.

$$F1 - Score = \left( 2 * \frac{Precision \;*\; Recall}{Precision \;+\; Recall} \right) \qquad (4)$$

## RESULTS

This section presents the performance of the proposed ensemble model for stroke prediction utilizing the Imbalance and Balanced Stroke datasets. In addition, the performance of all nine ML classifiers is compared with the proposed model. Also, sampling techniques, for example, ADASYN and SMOTE, are compared to evaluate the results of AI-based machine learning classifiers.

### Performance evaluation of single ML classifiers

Machine learning models, referred to as classifiers, have the ability to identify patterns within unseen data and provide better predictions. The dynamic nature of these models allows for adaptation over time as new data is included, in contrast to rule based-models that need explicit coding. Machine learning models may be classified into two main types: supervised and unsupervised. The primary differentiation between the two approaches lies in the fact that an unsupervised model is capable of processing unprocessed, unlabeled datasets, whereas a supervised technique needs labeled input and output training data. This study employed supervised technique. There are several methods available for evaluating a classification model. The most commonly utilized metric is accuracy. The other metrics were also utilized to effectively predict the classification results. Table 3 presents the prediction performance of all nine mostly used ML classifiers using an imbalanced dataset. The RF model achieved 95.8% highest prediction accuracy with an imbalanced dataset, and the SVM achieved the lowest 92.3% accuracy score overall. The RF model achieved 1,470 true positives (TP), which is higher than other ML models with an imbalanced dataset. DT achieved 1,402 true positives (TP). SVM performs worst with 118 wrong predictions, and LR performs best with only 63 wrong predictions, according to the confusion matrix values presented in Table 3.

The efficiency of ML classifiers based on the SMOTE oversampling technique is presented in Table 4. Different ML classifiers, for example, SVM, DT, ETC, KNN, SGD, RF, GBM, LR, and ADA, are evaluated with balanced data. The best-performing classifier is RF on a balanced dataset with 97.6% accuracy, whereas the worst-performing classifier is SVM with 62.8% accuracy. Table 4 shows that SVM also performs worst in the balanced dataset case, as shown in Table 3, where SVM achieved the lowest accuracy. The four well-

**Table 3 Evaluation of ML classifiers using imabalanced dataset.**

| Classifiers | Accuracy | Precision | Recall | F1 score | TP | FP | TN | FN | CP | WP |
|---|---|---|---|---|---|---|---|---|---|---|
| SVM | 92.3 | 92 | 92 | 92 | 1,410 | 60 | 5 | 58 | 1,415 | 118 |
| DT | 92.4 | 93 | 92 | 93 | 1,402 | 68 | 15 | 48 | 1,417 | 116 |
| ETC | 95.6 | 92 | 96 | 94 | 1,466 | 4 | 0 | 63 | 1,466 | 67 |
| KNN | 93.8 | 92 | 94 | 93 | 1,439 | 31 | 0 | 63 | 1,439 | 94 |
| GBM | 93.2 | 94 | 93 | 93 | 1,415 | 55 | 14 | 49 | 1,429 | 104 |
| SGD | 93.5 | 92 | 94 | 93 | 1,432 | 38 | 2 | 61 | 1,434 | 99 |
| RF | 95.5 | 92 | 95 | 95 | 1,464 | 6 | 0 | 63 | 1,464 | 69 |
| LR | 95.8 | 92 | 96 | 94 | 1,470 | 0 | 0 | 63 | 1,470 | 63 |
| ADA | 95.4 | 93 | 95 | 94 | 1,461 | 9 | 2 | 61 | 1,463 | 70 |

**Table 4 Evaluation of ML classifiers using SMOTE oversampling technique.**

| Classifiers | Accuracy | Precision | Recall | F1 score | TP | FP | TN | FN | CP | WP |
|---|---|---|---|---|---|---|---|---|---|---|
| SVM | 62.8 | 63 | 63 | 63 | 945 | 547 | 888 | 537 | 1,833 | 1,084 |
| DT | 97.1 | 97 | 97 | 97 | 1,409 | 83 | 1,425 | 0 | 2,834 | 83 |
| ETC | 97.2 | 97 | 97 | 97 | 1,412 | 80 | 1,425 | 0 | 2,837 | 80 |
| KNN | 94.5 | 95 | 95 | 95 | 1,334 | 158 | 1,425 | 0 | 2,759 | 158 |
| GBM | 97.1 | 97 | 97 | 97 | 1,410 | 82 | 1,425 | 0 | 2,835 | 82 |
| SGD | 66.2 | 67 | 66 | 66 | 1,176 | 316 | 757 | 668 | 1,933 | 984 |
| RF | 97.6 | 98 | 98 | 98 | 1,422 | 70 | 1,425 | 0 | 2,847 | 70 |
| LR | 78.1 | 79 | 78 | 78 | 1,087 | 405 | 1,192 | 233 | 2,279 | 638 |
| ADA | 80.9 | 81 | 82 | 82 | 1,089 | 403 | 1,271 | 154 | 2,360 | 557 |

known ML classifiers (DT, ETC, GBM, and RF) achieved 97% accuracy, but other classifiers' accuracy is low.

Table 5 depicts the performance of ML classifiers based on the ADASYN oversampling technique. Using balanced data, several ML classifiers, including SVM, DT, ETC, KNN, SGD, RF, GBM, LR, and ADA, are compared. On a balanced dataset, ETC is the most accurate classifier with 94.1% accuracy, while SVM is a less precise classifier with 64.5% accuracy. As shown in Table 4, SVM also performs inadequately in the case of a balanced data set, where it achieved the lowest accuracy. TWO well-known ML classifiers (RF, ETC) achieved a 93.7% and 94.1% accuracy, respectively while the accuracy of other classifiers is low.

The frequency of correct and wrong predictions is represented using count values and disaggregated by individual classes. Correct (accurate) predictions are achieved by summing the true positive (TP) and true negative (TN) values, whereas wrong (inaccurate) predictions are acquired by summing the false positive (FP) and false negative (FN) values. Figure 5 provides the Total predictions, correct predictions (CP), and wrong predictions (WP) made by the ML models. Figure 5A shows that SVM produced 118 wrong

**Table 5 Evaluation of ML classifiers using ADASYN oversampling technique.**

| Classifiers | Accuracy | Precision | Recall | F1 score | TP | FP | TN | FN | CP | WP |
|---|---|---|---|---|---|---|---|---|---|---|
| SVM | 64.5 | 65 | 65 | 65 | 958 | 533 | 928 | 503 | 1,886 | 1,036 |
| DT | 92.1 | 93 | 92 | 92 | 1,362 | 129 | 1,332 | 99 | 2,694 | 228 |
| ETC | 94.1 | 94 | 94 | 94 | 1,379 | 112 | 1,373 | 58 | 2,752 | 170 |
| KNN | 92.2 | 92 | 93 | 93 | 1,270 | 221 | 1,423 | 8 | 2,693 | 229 |
| GBM | 92.5 | 93 | 93 | 93 | 1,367 | 124 | 1,338 | 93 | 2,705 | 217 |
| SGD | 72.6 | 74 | 73 | 72 | 928 | 563 | 1,196 | 235 | 2,124 | 798 |
| RF | 93.7 | 94 | 94 | 94 | 1,371 | 120 | 1,367 | 64 | 2,738 | 184 |
| LR | 77.9 | 78 | 78 | 78 | 1,097 | 394 | 1,180 | 251 | 2,277 | 645 |
| ADA | 88.3 | 88 | 88 | 88 | 1,299 | 192 | 1,280 | 151 | 2,579 | 343 |

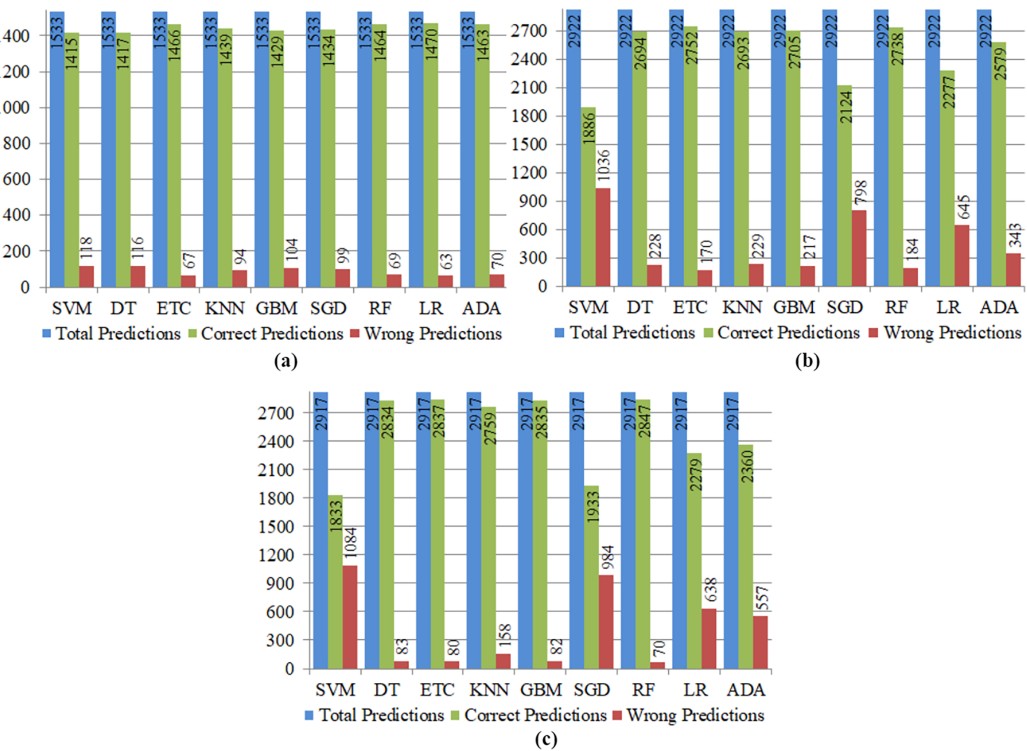

**Figure 5** Total correct and wrong predictions produced by single ML classifiers for stroke using (A) predictions with imbalanced dataset, (B) predictions with ADASYN oversampling technique, (C) predictions with SMOTE technique.

predictions (WP) and 1,415 correct predictions (CP) with an imbalanced dataset. The SVM classifier has worse performance than any other classifier. LR produced 1,470 highest number of predictions that are correct. Figure 5B shows that SVM with the ADASYN Oversampling technique achieved high number of wrong predictions (WP). In Fig. 5C, we see SVM with the maximum number of wrong predictions (WP) and RF with the maximum number of correct- predictions (CP). These predictions demonstrate that

**Table 6 Evaluation of proposed RDET stacking classifier with K-FOLD.**

| Technique | Classifiers | Accuracy | Precision | Recall | F1 score | TP | FP | TN | FN | CP | WP |
|-----------|-------------|----------|-----------|--------|----------|------|----|------|----|-------|-----|
| Original dataset | Stacking ensemble | 95.3 | 92 | 95 | 94 | 1,462 | 8 | 0 | 63 | 1,462 | 71 |
| | Voting ensemble | 95.7 | 93 | 95 | 95 | 1,456 | 14 | 3 | 60 | 1,459 | 74 |
| SMOTE oversampled | Stacking ensemble | 100 | 100 | 100 | 100 | 1,491 | 1 | 1,425 | 0 | 2,916 | 1 |
| | Voting ensemble | 98.9 | 99 | 98 | 99 | 1,460 | 32 | 1,425 | 0 | 2,885 | 32 |
| ADASYN oversampled | Stacking ensemble | 96.4 | 96 | 97 | 96 | 1,433 | 58 | 1,382 | 49 | 2,815 | 107 |
| | Voting ensemble | 95.9 | 96 | 96 | 96 | 1,415 | 76 | 1,387 | 44 | 2,802 | 120 |

Balanced stroke data with SMOTE technique achieved high number of correct prediction (CP) and proved helpful to enhance the prediction performance.

## Performance evaluation of ensemble ML classifiers

This subsection presents the experimental results of Ensemble machine learning classifiers. The Ensemble of RF, DT, ETC is created with stacking and the voting classifier technique. Stacking is based on employing the best features of various models while discovering ways to combine their predictions to enhance overall accuracy effectively. Decision tree (DT), random forests (RFs) and extra tree classifier (ETC) can be stacked to produce a strong ensemble that combines the important features of these classifiers. ETC delt with extra-trees and captured intricate decision boundaries, whereas RFs flourished at handling nonlinear connections, missing data, and outliers.

K-fold validation is a statistical methodology employed to evaluate the effectiveness of machine learning models. The utilization of this approach is prevalent in the field of supervised machine learning for the purpose of evaluating and choosing a model suitable for a specific task. This is due to its inherent simplicity in understanding, ease of implementation, and ability to provide skill estimates that are generally less biased compared to alternative methodologies. The number of groups into which a given data sample is to be partitioned is determined by a single parameter called k. Therefore, the methodology is well recognized as K-Fold. When a particular value is chosen for the variable K, it may be utilized in model, resulting in K = 10 being represented as 10-Fold cross validation.

Table 6 evaluates the proposed RDET classifier results with four performance metrics using stacking and soft voting ensemble classifiers. On original (Imbalanced dataset), both techniques for Ensembling the ML classifiers achieved above 95% accuracy. Using ADASYN technique, Stacking Ensemble achieved 96.4% accuracy and 97% recall score, while Voting classifier using the same technique achieved 95.9% accuracy. Figure 6 illustrate the correct and wrong predictions produced by Ensemble ML Classifiers for stroke using the Original and Balanced dataset with SMOTE, and ADADYN technique.

The T-test is employed to assess the level of statistical differentiation between the models. The utilization of this approach is commonly observed in hypothesis testing, whereby its purpose is to evaluate the impact of a particular method on the target population or to determine the differences between several models. In the statistical test
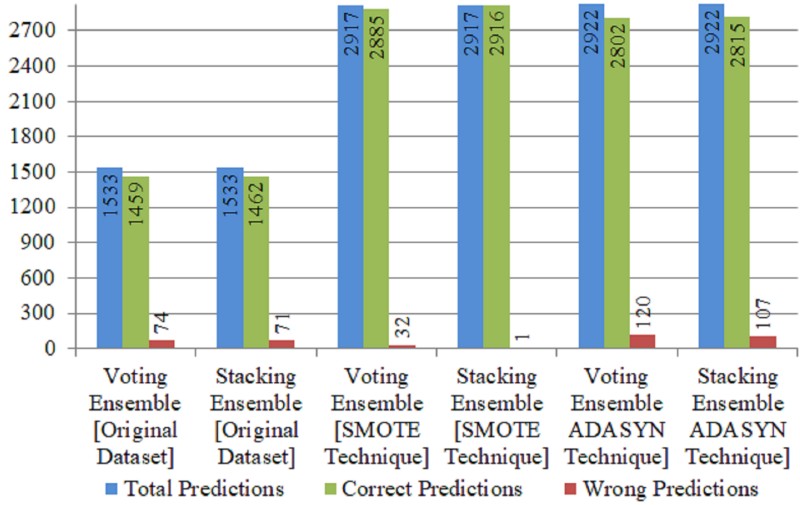

**Figure 6 Correct and wrong predictions produced by ensemble ML classifiers for stroke using original and balanced dataset with SMOTE, and ADADYN technique.**

**Table 7 Evaluation of the proposed method *vs.* other models using T-test.**

| Model | *P* value | S.T | Hypothesis |
|---|---|---|---|
| *vs.* SVM | 0.0000 | −740.9999 | Rejected |
| *vs.* DT | 0.0000 | −119.0000 | Rejected |
| *vs.* ETC | 0.0000 | −58.9999 | Rejected |
| *vs.* KNN | 0.0000 | −41.0000 | Rejected |
| *vs.* GBM | 0.0000 | −119.0000 | Rejected |
| *vs.* SGD | 0.0000 | −141.5683 | Rejected |
| *vs.* RF | 0.0002 | −20.9999 | Rejected |
| *vs.* LR | 0.0000 | −89.4720 | Rejected |
| Proposed stack *vs.* voting | 0.0134 | 5.2509 | Rejected |
| Proposed original *vs.* SMOTE | 0.0135 | −5.2344 | Rejected |
| Proposed sacking ensemble (SMOTE) *vs.* sacking ensemble (ADASYN) | 0.0005 | 15.4470 | Rejected |

known as the T test, distinct scenarios are taken into consideration, as shown in Table 7. The t-test is utilized to determine the statistical significance of one strategy relative to another by either accepting or rejecting the null-hypothesis. This study adapted two scenarios; null hypothesis: $> \mu1 = \mu2$, alternative hypothesis: $> \mu1 \neq \mu2$. In the first scenario, the population demonstrates that the results of the proposed approach compared to those of the comparative methodology are equal. The results that were observed do not indicate any statistical significance. In the second scenario, the population demonstrates that the results of the proposed approach compared to those of the comparative methodology are not equal. The obtained results suggest statistical significance.

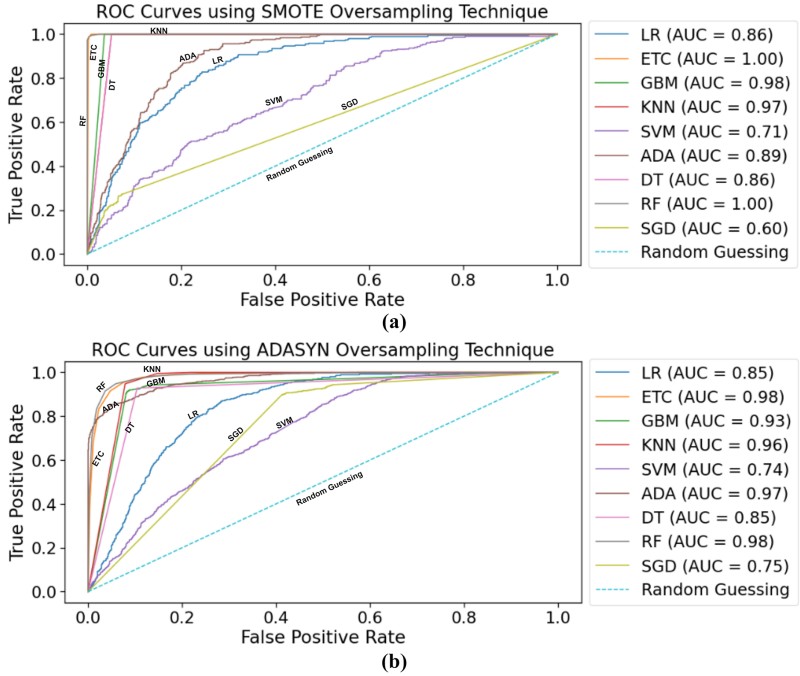

**Figure 7 Samples taken from the red blood cell image datasets contain parasitized cell images and uninfected cell images.**

## ROC curves

The most straightforward approach for visualizing the accuracy and loss training and testing curves is through the use of the receiver operating characteristic—area under the curve (ROC-AUC) metric. The ROC curve, also known as the receiver operating characteristic curve, is a graphical representation that illustrates the performance of a classification model over various classifi criteria. The shown curve represents two parameters, namely the true positive rate and the false positive rate.

Figure 7A indicates the area under the ROC curves of ML classifiers using a balanced dataset with the SMOTE oversampling technique. Both supervised ML classifiers RF, ETC, achieved a 1.00 AUC score. The true positive rate is the highest of these two classifiers. Other classifiers, such as LR and DT, achieved the same 0.85 AUC-score, whereas SGD achieved the lowest 0.68 AUC-score. Overall, the performance of all ML classifiers seems very good.

The ROC curves for ML classifiers using the ADASYN oversampling technique are depicted in Fig. 7B. As with using SMOTE for balancing the dataset, RF, ETC achieved a 0.98 AUC score, which is the highest. After this, the ADA boosting classifier achieved a 0.97 AUC-score for predicting stroke risk. SVM needs to achieve better results using the ADASYN technique. The lowest AUC score achieved by the SVM classifier is 0.71.

**Table 8 Comparison of stacking and voting ensemble models with previous study.**

| Authors | Methods | Accuracy | Precision | Recall | F1 score | Year |
|---|---|---|---|---|---|---|
| Bandi, Bhattacharyya & Midhunchakkravarthy (2020) | Improvised RF | 96.97 | 94.56 | 94.9 | 94.73 | 2020 |
| Alruily et al. (2023) | RXLM | 96.34 | 96.12 | 96.55 | 96.33 | 2023 |
| Govindarajan et al. (2020) | ANN | 95.3 | 95.9 | 95.9 | – | 2020 |
| Monteiro et al. (2018) | RF | 92.6 | – | – | – | 2018 |
| Li et al. (2019) | RF | 97.07 | 97.33 | 98.44 | 97.88 | 2019 |
| Sailasya & Kumari (2021) | NB | 82 | 79 | 85 | 82 | 2021 |
| Nwosu et al. (2019) | Neural network | 75.02 | – | – | – | 2019 |
| Ong et al. (2020) | RNN | 89 | 93 | 90 | 87 | 2020 |
| Tazin et al. (2021) | RF | 96 | 96 | 96 | 96 | 2021 |
|  | Stacking ensemble | 100 | 100 | 100 | 100 | 2023 |
|  | Voting ensemble | 98.9 | 99 | 98 | 99 | 2023 |

## Comparison of proposed RDET stacked classifier with state of the art study

We compared our proposed RFET Stacked Classifier results with state-of-the-art studies to validate its efficacy and robustness. Table 8 demonstrates the comparison results of various studies. *Bandi, Bhattacharyya & Midhunchakkravarthy (2020)* adopted an improvised random-forest RF method to predict stroke and achieved 96.9% accuracy. A article (*Alruily et al., 2023*) used an ensemble of random forest, XG-Boost, and lightweight models in 2023 and achieved 96.34% accuracy and a 96.5% recall score. In 2020, authors developed artificial neural networks (*Govindarajan et al., 2020*) that work with 95.3% accuracy in stroke prediction. The authors did not compute an F1 score in this study. Also, this study has limited performance. *Monteiro et al. (2018)* and *Li et al. (2019)*, both authors, presented the RF method for stroke prediction. However, *Monteiro et al. (2018)* achieved 92.6% low results because they measured the performance only with one metric accuracy. Other metrics are ignored, which are significant for assessing performance. Likewise, *Govindarajan et al. (2020)* and *Nwosu et al. (2019)*, used neural networks, but their accuracy was worse. They did not perform any other comparisons or use performance metrics effectively. *Tazin et al. (2021)* also utilized an RF model in 2021 with a 96% accuracy rate. All the state-of-the-art studies discussed in the literature have low accuracy and imbalanced dataset issues, and the proposed methods needed to be more accurate to detect stroke early. Thus, our proposed approach is more Efficient and robust than previous methods in detecting strokes from a balanced dataset with high performance.

## CONCLUSION

In this study, we propose a stacking ensemble ML classifier for stroke prediction and address the class Imbalance issues with the most important oversampling techniques. In medical disease diagnosis, machine learning has made a significant contribution towards the early-prediction of strokes and reducing their severe aftereffects. This study utilized various ML classifiers with fine-tuned parameters to effectively predict the stroke. The

random forest (RF) classifier attained a 97% accuracy score in the balanced stroke prediction dataset. To enhance the classifier's performance and reduce overfitting issues, we propose a stacking ensemble classifier, that is more reliable and efficient for predicting stroke disease. The study employed three tree based Ml classifiers to make a final stronger classifier. The random forest (RF), decision tree (DT), and extra tree classifier (ETC) are first trained and fine-tuned with hyperparameters. Then combined through a stacking classifier with a k-fold cross validation methodology. The Ensemble Stacking classifier makes predictions on new or unseen data. The effectiveness of the proposed ensemble classifier is evaluated through various performance metrics. The proposed approach has an extreme ability to predict the stroke with 100% accuracy. The proposed approach makes 2,916 accurate predictions (correct predictions) out of a total of 2,917 predictions. The stacking ensemble classifier makes an exceptional contribution to stroke prediction and suggests superior efficacy compared to single ML models. To recover from stroke and maintain social connections, the proposed voting ensemble method accurately and early predicted the stroke with minimal errors. The proposed approach presents great significance in addressing the risk of stroke since individuals experiencing memory impairments face challenges in cognitive functioning and decision-making abilities in a social context. In future studies, we may combine different stroke prediction datasets or collect more data about brain stroke. We developed a more reliable feature extractor classifier and processed the data with deep learning, then checked the results on new large data sets.

### Funding
This research was funded by Princess Nourah bint Abdulrahman University and Researchers Supporting Project number (PNURSP2023R346), and Princess Nourah bint Abdulrahman University, Riyadh, Saudi Arabia. The Prince Sultan University, Riyadh Saudi Arabia supported the APC of this publication. The funders had a role in the study design, data collection and analysis, decision to publish, or preparation of the manuscript.

### Grant Disclosures
The following grant information was disclosed by the authors:
Princess Nourah bint Abdulrahman University: PNURSP2023R346.
Princess Nourah bint Abdulrahman University, Riyadh, Saudi Arabia.

### Competing Interests
The authors declare that they have no competing interests.

### Author Contributions
- Amjad Rehman conceived and designed the experiments, performed the experiments, analyzed the data, prepared figures and/or tables, and approved the final draft.
- Teg Alam analyzed the data, performed the computation work, authored or reviewed drafts of the article, and approved the final draft.

- Muhammad Mujahid conceived and designed the experiments, performed the experiments, analyzed the data, performed the computation work, prepared figures and/or tables, authored or reviewed drafts of the article, and approved the final draft.
- Faten S. Alamri analyzed the data, prepared figures and/or tables, authored or reviewed drafts of the article, and approved the final draft.
- Bayan Al Ghofaily performed the computation work, prepared figures and/or tables, authored or reviewed drafts of the article, and approved the final draft.
- Tanzila Saba conceived and designed the experiments, performed the experiments, analyzed the data, performed the computation work, authored or reviewed drafts of the article, and approved the final draft.

## Data Availability

The implementation code and the Stroke Prediction Dataset areavailable in the Supplemental File. The dataset is also available at Kaggle: https://www.kaggle.com/datasets/fedesoriano/stroke-prediction-dataset.

## Supplemental Information

Supplemental information for this article can be found online at http://dx.doi.org/10.7717/peerj-cs.1684#supplemental-information.

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
