# Peer review of "RDET stacking classifier: a novel machine learning based approach for stroke prediction using imbalance data"

_PeerJ Computer Science, doi:10.7717/peerj-cs.1684_

## Round 0.1 · original submission · Major Revisions

Please follow the reviewers' comments.

**Language Note:** PeerJ staff have identified that the English language needs to be improved. When you prepare your next revision, please either (i) have a colleague who is proficient in English and familiar with the subject matter review your manuscript, or (ii) contact a professional editing service to review your manuscript. PeerJ can provide language editing services - you can contact us at copyediting@peerj.com for pricing (be sure to provide your manuscript number and title). – PeerJ Staff

Reviewer 1 ·

Basic reporting

Need to Improve the Literature.

Experimental design

Adequate and sufficient

Validity of the findings

Adequate and sufficient

Additional comments

Title: RDET Stacking classifier: A Novel Machine 2 Learning based approach for Stroke prediction using imbalance data
I congratulate the authors for their contribution and hard work. Before going to submit the publish the manuscript, a few perhaps would get clarified from my end.
1. In this work, the authors utilized the metrics such as accuracy, precision, recall, and F1-score. In abstract, the authors proposed nine ML classifiers with Hyper-parameter tuning to predict the stroke and compare the effectiveness of Proposed approach with other classifiers and the authors mentioned that the “experimental outcomes demonstrated the superior performance of the stacking classification method compared to other approaches. The stacking method achieved a remarkable accuracy of 100% as well as exceptional F1-Score, precision, and recall score”. Elaborate with more clarifications.
2. Add more recent references (2021, 2022, 2023).
3. The paper is technically sound but need to improve the grammar.
4. Provide more relevant information regarding the tables and figures.
5. What is the significance of the work towards Societal Context.

Reviewer 2 ·

Basic reporting

This paper used an ensemble of machine learning models using voting classifiers to predict the stroke using imbalanced data. Extensive experiments were conducted by the authors to demonstrate the efficacy of the proposed method, with outstanding results. The topic is interesting and valuable for the scientific community. However, some minor points should be resolved before publication.
• The introduction section should be improved with the latest published articles related to stroke prediction.
• The related work section is missing a comparison table that presents the pros and cons.
• Please add the hyperparameters of utilized machine learning models in the table form in methodology section.
• Some text should be added in the conclusion section about future work.

Experimental design

I am satisfied with the experimental design

Validity of the findings

See the above comments.

Additional comments

NA

·

Basic reporting

- Introduction
-The introduction section is too general, and it introduces concepts that are well known about . This work focuses on proposing and assessing A stroke is a sudden neurological condition affecting the blood vessels in the brain. It arises when the blood flow to a specific brain region is interrupted, leading to oxygen deprivation to the brain cells. The cerebrum, a vital central nervous system component, is responsible for governing various physiological processes including memory, movement, cognition, speech, and the autonomic regulation of essential organs. Nevertheless, the outcomes can be potentially life-threatening.. Furthermore, "the research motivation" at the introduction section is missing. Please rewrite this section.

Experimental design

- Related work
-In this section, the authors should be describe some of the research works about the most recent progress made in harnessing the influence of Machine learning (ML) methods to forecast the risk of stroke. Particularly noteworthy is the successful integration of ML approaches, which have demonstrated significant potential in providing more precise predictions of stroke outcomes when compared to conventional methods.. Author’s are advised to include more latest work in his research paper.

Validity of the findings

-In addition, a conclusion of related work in the forms of a table in terms of utilized technique, evaluation tools, data set, performance metrics, advantages, and disadvantages could reconcile from other researchers work to the own one.
-Model building & Experimental Setup.
-Please provide a sequence diagram to show the interaction between components of the proposed architecture according to Figures 1. Kindly reshape all the figures in high dpi.

-Please provided real-world case study example for better understanding the proposed approach in more details.
Conclusion should be improved in terms of Stacking Ensemble ML classifier for Stroke prediction

Additional comments

The evaluation is incomplete. I would like to see an evaluation on the proposed solution in terms of Stacking Ensemble ML classifier for Stroke prediction obtaining more realistic models.
-Paper needs some revision in English. The overall paper should be carefully revised with focus on the language: especially grammar and punctuation.
-Overall, there are still some major parts that the authors did not explain clearly. Some additional evaluations are expected to be in the manuscript as well. As a result, I am going to suggest Major revision of the paper in its present form.

---

## Round 0.2 · accepted · Accept

The author followed all review recommendations.

Reviewer 1 ·

Basic reporting

The authors provided the adequate information to all my Queries.

Experimental design

Adequate and Sufficient

Validity of the findings

Adequate and sufficient

Additional comments

Nil

·

Basic reporting

Clear

Experimental design

Good

Validity of the findings

Correct

Additional comments

Thanks to the authors for the detailed response and additions. I read through the comments and skimmed the revised PDF, and the updates significantly improved the paper. I would be happy to recommend this paper for publication.